# Structural insight into catalytic mechanism of PET hydrolase

Xu Han[1], Weidong Liu[1], Jian-Wen Huang[2], Jiantao Ma[1,3], Yingying Zheng[1], Tzu-Ping Ko[4], Limin Xu[2], Ya-Shan Cheng[2], Chun-Chi Chen[1] & Rey-Ting Guo[1]

PET hydrolase (PETase), which hydrolyzes polyethylene terephthalate (PET) into soluble building blocks, provides an attractive avenue for the bioconversion of plastics. Here we present the structures of a novel PETase from the PET-consuming microbe *Ideonella sakaiensis* in complex with substrate and product analogs. Through structural analyses, mutagenesis, and activity measurements, a substrate-binding mode is proposed, and several features critical for catalysis are elucidated.

[1] Industrial Enzymes National Engineering Laboratory, Tianjin Institute of Industrial Biotechnology, Chinese Academy of Sciences, Tianjin 300308, China. [2] AsiaPac Biotechnology Co., Ltd, Dongguan 523808, China. [3] School of Biotechnology, Jiangnan University, Wuxi 214122, China. [4] Institute of Biological Chemistry, Academia Sinica, Taipei 11529, Taiwan. Xu Han, Weidong Liu and Jian-Wen Huang contributed equally to this work. Correspondence and requests for materials should be addressed to R.-T.G. (email: guo_rt@tib.cas.cn)

Polyethylene terephthalate (PET) is high molecular weight polymer composed of ester bond-linked terephthalate (TPA) and ethylene glycol (EG)[1]. The durability and other favorable physical properties made PET one of the most extensively utilized plastic. However, the large amounts of PET that enters and accumulates in the ecosystem pose a great environmental challenge.

PET-hydrolyzing enzymes that decompose PET into its building blocks could provide an eco-friendly solution to PET accumulation in the environment[2]. Hydrolases, including esterases, lipases, and cutinases[3–7] have previously been reported to exhibit PET-degrading activity. The catalytic mechanism of cutinase, which belongs to the serine hydrolase family, has been explored by analyzing complex structures of inhibitors[8, 9] and natural substrate analogs (cutin and triglyceride, Fig. 1)[10–12]. However, PET is very distinct from cutinase substrates in terms of chemical structures, and the precise PET-binding mode and underlying mechanism of PET hydrolysis remain unclear.

Recently, a novel bacterium *Ideonella sakaiensis* 201-F6 was isolated, which could utilize PET as an energy and carbon source[13]. Specifically, *I. sakaiensis* adheres to PET surface and secretes a unique cutinase-like enzyme that degrades PET. By hydrolyzing PET, the enzyme produces mono-(2-hydroxyethyl) terephthalic acid (MHET), TPA and bis-2(hydroxyethyl) TPA (BHET) (Fig. 1). Notably, the enzyme is highly homologous to several cutinases with PET-hydrolyzing activity but exhibits lower activity on *p*-nitrophenol-linked aliphatic esters (Supplementary Fig. 1), which are the preferred substrates for lipases and cutinases. Instead, it exhibits 5.5- to 120-fold higher activity against PET over the other enzymes. The enzyme is named PET hydrolase (PETase) for its PET preference. PETase displays outstanding performance in hydrolyzing PET and holds a great potential in further applications. In this study, in order to determine the mechanism of enzymatic PET hydrolysis, we have determined the crystal structures of native PETase and two ligand-bound complexes.

## Results

**Overall structures of PETase**. The recombinant protein of wild-type PETase without signal peptide was expressed and crystallized in the orthorhombic space group $P2_12_12_1$ (Supplementary Table 2). The structure was solved at 1.58 Å resolution, and three polypeptide chains were observed in an asymmetric unit, which are denoted A, B, and C chain. PETase adopts the canonical α/β-hydrolase fold, in which the strictly conserved catalytic triad S131-H208-D177, as predicted from sequence alignment with homologous enzymes that exhibit PET-hydrolytic activity (Supplementary Fig. 1), is found on the protein surface (Fig. 2a). S131 serves as the nucleophile and is located within hydrogen-bond distance to be polarized by the base H208, which is in turn stabilized by the acid D177 (Fig. 2b). Several unique features are

**Fig. 1** Structures of PETase hydrolysis products (boxed) and other compounds used in this study

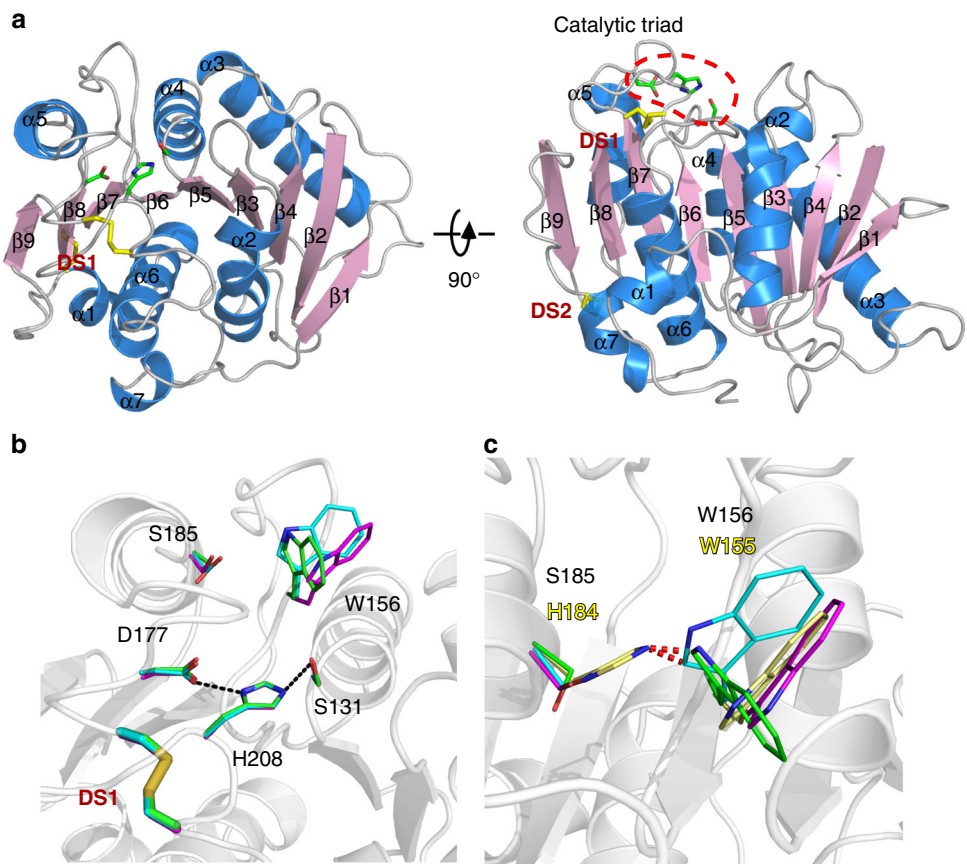

**Fig. 2** Overall structure of apo-form PETase. **a** The PETase structure is presented as a cartoon model. The catalytic triad (red dash line circle, top left) and disulfide bridges (red labels) are shown as sticks. **b** Three polypeptide chains in an asymmetric unit are superimposed. W156, S185 and the catalytic triad of A (green), B (cyan), and C (magenta) chain are shown in the foreground. Dash lines indicate distances <3.5 Å. **c** The PETase in this study is compared with the *Thermobifida fusca* PET degrading hydrolase (PDB ID, 4CG2). The corresponding residues of W156 and S185 in the latter enzyme are colored yellow. Red dash lines indicate distances <1.5 Å

present near the catalytic center. First, PETase forms two intramolecular disulfide bridges (DS1 and DS2) while the other homologous enzymes have only one (Supplementary Figs. 1 and 2). The strictly conserved DS2 (C244-C260) connects the C-terminal helix and the last loop (Supplementary Fig. 2), while the PETase-specific DS1 (C174-C210) links two loops that harbor the catalytic acid and base (Fig. 2b; Supplementary Fig. 1). Second, W156 adjacent to the catalytic center displays variable conformations in the crystal structures (Fig. 2b). Interestingly, an equivalent Trp is found in every PET-hydrolyzing enzyme, but W156 wobbling has not been observed before (Fig. 2c; Supplementary Fig. 3). In fact, this Trp in the other structures all adopted the "C" conformer (Fig. 2c; Supplementary Fig. 3). Among the surrounding amino acids a distinct S185 is found in PETase, which is consistently replaced by a His in the homologous enzymes (Fig. 2c; Supplementary Figs. 1 and 3). Further analyses indicate that a His side chain would be too close to allow "A" or "B" conformer of W156 in PETase, while a Ser residue can yield a space to accommodate the "C" conformer of W156 (Fig. 2c). Therefore, the W156 wobbling is due to the presence of S185.

**Complex structures of PETase**. Next, we tried to obtain complex structures by co-crystallizing or soaking the wild-type PETase crystals with various ligands. However, no additional electron density corresponding to ligand could be observed. We then used the inactive variant protein S131A to prevent ligand hydrolysis. Indeed, S131A variant showed very low activity comparing to the wild-type protein (Fig. 3c). However, attempts to obtain complex structures using the S131A variant, which crystallized in a C2 unit cell (Supplementary Table 2), also failed. In both crystal forms, the R103 side chain of one protein molecule penetrates into a surface cleft next to S131 (or A131) of another protein molecule (Supplementary Fig. 4). It appeared likely that the R103 side chain occupied the putative substrate-binding groove and prevented the substrate from entering. The wild type and S131A proteins also failed to crystallize in a different unit cell by screening for many other crystallization conditions. We therefore produced and crystallized the double mutant R103G/S131A (DM) for complex formations. A comparison of the wild type, S131A, and DM crystal structures suggests that these mutations did not significantly alter the overall protein fold (Cα root mean square deviation = 0.144–0.2 Å). Soaking the DM crystals with 1-(2-hydroxyethyl) 4-methyl terephthalate (HEMT) and *p*-nitrophenol (pNP) produced clear densities for both ligands, and each compound could be modeled into a shallow cleft on the protein surface (Supplementary Fig. 5).

In DM-HEMT, the ligand is bound in a groove containing A131, H208, W156, I179, W130, Y58, and M132 (Fig. 3a). The H208 side chain is within H-bond distance to the ester O atom of HEMT, W156 that adopts the "B" conformer is T-stacked to the benzene ring, and the other residues provide hydrophobic interactions. The proximal carbonyl group lies adjacent to

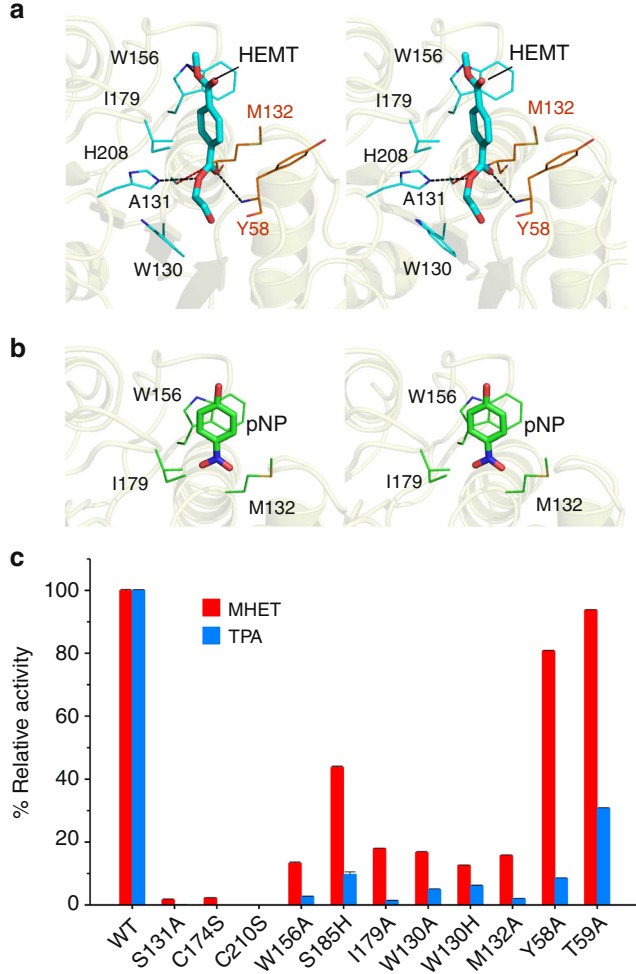

**Fig. 3** Substrate-binding residues and the variants. The detailed enzyme–ligand interaction networks in **a** DM-HEMT and **b** DM-pNP are shown in stereo view. The ligands and protein residues are presented as thick and thin sticks, respectively. The oxyanion hole-forming residues are in orange. Distances <3.5 Å are indicated with dash lines. In **c**, the production levels of MHET and TPA by each protein are presented as percentages of the wild type. Each measurement was conducted in triplicate, from which the average ± s.d. was calculated

A131, presumably facilitating nucleophilic attack by S131 in the wild-type enzyme. The backbone NH groups of M132 and Y58 that constitute an oxyanion hole, a common feature of other superfamily members including esterase and lipases[14], in the catalytic center are within H-bonding distance to the carbonyl O atom of HEMT (Fig. 3a). Judged by its binding mode, the HEMT in the complex behaves like a substrate analog. Therefore, a fragment of PET that consists of two terephthalate molecules linked by an ethylene glycol was docked into the enzyme based on the DM-HEMT structure (Supplementary Fig. 6a). The two phenyl groups are located on either side of A131 (or the catalytic S131) with a kink of about 120°. In the binding region for the 1st TPA unit, the same HEMT-interacting residues may bind to PET (Supplementary Fig. 6b). In comparison, fewer residues are found to contact the 2nd TPA unit. Only T59 and W130 might provide H-bonding and T-stacking interactions with the ester oxygen and the benzene ring (Supplementary Fig. 6b). Sequence alignment and structure superimposition indicate that the substrate-binding residues are strictly or semi-conserved among the PET-hydrolyzing enzymes (Supplementary Figs. 1 and 7). Thus it appears that PETase and the other enzymes with

PET-hydrolyzing activity use a similar substrate-interaction network.

The ligand of DM-pNP is located to the 1st benzene ring-binding site, the same side as HEMT. Because the nitro group of pNP is structurally similar to the carboxyl group of PETase products (TPA and MHET, see Fig. 1), the observed pNP binding mode in the complex may be analogous to the product. Fewer ligand interactions are found in DM-pNP than DM-HEMT that involve only hydrophobic contacts with W156, I179, and M132 (Fig. 3b). Notably, the locations of HEMT and pNP vary slightly. A comparison of the distance and angle between the benzene rings of the two ligands indicates that pNP is rotated by about 36° and further away from the active site by ~2.3 Å than HEMT (Supplementary Fig. 5b). As a result, the phenyl group of pNP becomes face-to-face stacked with the W156 indole (Fig. 3b).

**Activity analysis of PETase variants**. Based on the above structural analyses, additional variants were constructed and their PET hydrolyzing activity was measured (Fig. 3c). First, the DS1 disrupting variants C174S and C210S exhibited very low activity, suggesting that DS1 is essential for PETase activity. As the variant W156A showed low levels of MHET and TPA production (13.33 ± 0.13 and 2.77 ± 0.04%), W156 also appears to be an important residue for PETase activity. Notably, the conserved W156 residue shows wobbling conformation and only W156 in "B" conformation is favorable for HEMT binding (Supplementary Figure 8). As mentioned above, the W156 wobbling is allowed due to the presence of S185 in PETase but not in other homologous enzymes that possesses a His residue in the corresponding position. Therefore, the variant S185H was constructed and its activity measured. The resulting 43.87 ± 0.30% MHET and 9.73 ± 0.77% TPA production by S185H indicate that S185 has a role in PETase catalysis, and His substitution partially compromises activity. In addition, other HEMT-interacting residues including I179, W130, and M132 are also essential, as each corresponding Ala variant showed significantly lower activity. Notably, replacing W130 with a His, which is consistently observed in other homologous enzymes, significantly reduced the PETase activity. Y58, which also interacts with HEMT, appears to play a minimal role in hydrolyzing PET into MHET as Y58A exhibited 80.73% MHET production. However, because this variant is inefficient in TPA production (8.60%), Y58 should be essential in further MHET hydrolysis into TPA. A similar role is likely played by T59 because the variant T59A retained full activity in producing MHET but showed lower activity in producing TPA (30.90%).

By taking together all these results, a catalytic pathway of PETase can be proposed (Fig. 4). The apo-form enzyme provides a shallow cleft for substrate binding onto the protein surface, with the W156 side chain in various conformations. When the enzyme binds PET, the carbonyl group attached to the 1st benzene ring is directed toward the center of the substrate-binding cleft where the catalytic triad carries out nucleophilic attack and the oxyanion hole polarizes the ester bond and stabilizes the reaction intermediate. The subsequent formation of acyl-enzyme intermediate and second nucleophilic attack by a water molecule follow the classical mechanism of action of cutinase[15]. After the ester bond is cleaved, the remaining benzoic acid group of the product forms a broad planar surface which is readily face-to-face stacked with the W156 side chain. Thus the product is rotated and pulled away from the original, probably weaker, T-stacked position before it is released.

**Discussion**

In this study, we determined apo- and complex crystal structures of a novel PETase from a PET-consuming bacterium followed by

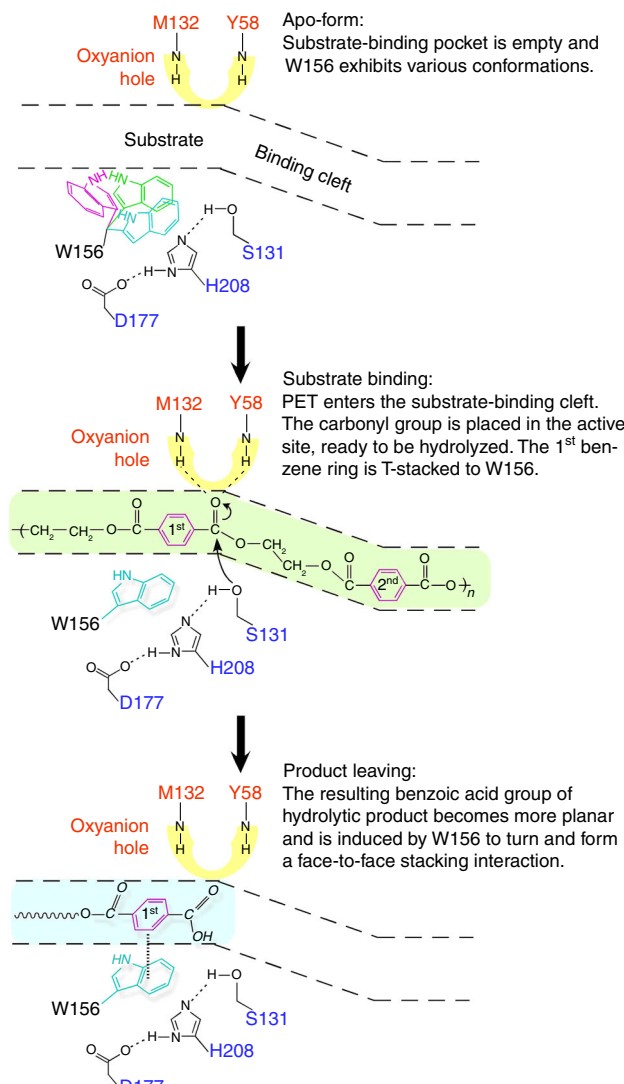

**Apo-form:**
Substrate-binding pocket is empty and W156 exhibits various conformations.

**Substrate binding:**
PET enters the substrate-binding cleft. The carbonyl group is placed in the active site, ready to be hydrolyzed. The 1st benzene ring is T-stacked to W156.

**Product leaving:**
The resulting benzoic acid group of hydrolytic product becomes more planar and is induced by W156 to turn and form a face-to-face stacking interaction.

**Fig. 4** Proposed catalytic mechanism of PETase

enzymatic assays to identify key residues requires for catalysis. PETase adopts a similar overall fold as related enzymes, but harbors several unique structural features near the active center. The enzyme possesses an additional disulfide bridge that links the loops that bear the catalytic acid and base. This disulfide bridge appears to play an important role in PETase function as activity is severely compromised once DS1 is disrupted. The structures with bound substrate and product analogs (HEMT and pNP) showed both ligands bound to a shallow cleft formed mainly by hydrophobic residues on the protein surface. The mutagenesis results indicate that these substrate-interacting residues are also important for the enzymatic activity. One of them is W156, whose conformation wobbles in the apo-form enzyme. It may facilitate substrate binding and product leaving. Notably, the W156 wobbling is coupled to S185. The corresponding residue of S185 in other homologous structures is a His, whose side chain prevents the equivalent Trp to W156 from adopting alternative conformations. The reduced activity of the variant S185H suggests that the outstanding performance of PETase dependents, at least partially, on the presence of S185. In conclusion, this study suggests substrate-binding mode in a PET-hydrolyzing enzyme, which can provide guidance for further engineering and applications of PETase in the bioconversion of plastics.

## Methods

**Cloning.** The PETase from *Ideonella sakaiensis* (GenBank accession number: GAP38373.1) without the N-terminal 29 amino acids was chemically synthesized (GENE ray Biotech Co., Shanghai, China), ligated into the pET32a vector, and expressed in *Escherichia coli*.

**Site-directed mutagenesis.** Variants were constructed by using a QuickChange site-directed mutagenesis kit (Agilent Technologies, Santa Clara, CA, USA) with the wild-type PETase plasmid as a template. R103G/S131A (DM) was constructed by using S131A as the template. The sequences of the mutagenesis oligonucleotides are listed in Supplementary Table 1. The PCR products were incubated with DpnI (New England Biolabs, Hitchin, UK) to digest the original DNA template and then separately transformed into *E. coli* strain XL1-Blue. The mutations were confirmed by sequencing.

**Protein purification.** The pET32a-PETase plasmid, either wild type or variants, was transformed into *E. coli* BL21trxB(DE3) cells that were grown in LB medium at 37 °C to an $OD_{600}$ of ~0.8 and then induced by 0.6 mM isopropyl β-D-thiogalactopyranoside (IPTG) at 16 °C for 24 h. Cells were harvested by centrifugation at $5000 \times g$ for 15 min and then re-suspended in a lysis buffer containing 25 mM Tris-HCl, 150 mM NaCl and 20 mM imidazole (pH 7.5), followed by disruption with a French Press. Cell debris was removed by centrifugation at $17,000 \times g$ for 1 h. The supernatant was then applied to a Ni-NTA column with FPLC system (GE Healthcare). The target protein was eluted at ~70 mM imidazole when using a 20–250 mM imidazole gradient. The target protein containing fractions were pooled and dialyzed against a buffer containing 25 mM Tris-HCl and 150 mM NaCl (pH 7.5), and subjected to TEV protease digestion overnight to remove the His tag. The mixture was then passed through another Ni-NTA column. The untagged protein was eluted with the imidazole free buffer. The purity of each protein (>95%) was checked by SDS-PAGE analysis. The purified proteins were each concentrated to 30 mg mL$^{-1}$ for crystallization screening.

**Structure determination and refinement.** All crystallization experiments were conducted at 25 °C using the sitting-drop vapor-diffusion method. In general, 1 µL PETase containing solution (25 mM Tris-HCl, 150 mM NaCl, pH 7.5; 30 mg mL$^{-1}$) was mixed with 1 µL of reservoir solution in 48-well Cryschem Plates, and equilibrated against 100 µL of the reservoir solution. The optimized crystallization condition of PETase was 30% PEG6000 and 0.1 M MES (pH 6.0). The optimized crystallization condition for S131A was 1.2 M ammonium sulfate, 0.2 M NaCl, 0.1 M Bis-Tris (pH 6.5), and for DM it was 1.1 M ammonium sulfate, 0.1 M NaCl, 0.1 M Hepes (pH 7.5). Within 3–4 days, the crystals reached sizes suitable for X-ray diffraction. The DM crystals in complex with pNP and HEMT were obtained by soaking crystals with mother liquor containing 10 mM each compound for 50 h.

All X-ray diffraction data sets were collected at a wavelength of 1.00 Å on the beam line BL15A1 and TPS05A in the National Synchrotron Radiation Research Center (NSRRC) in Hsinchu, Taiwan. The crystals were mounted in a cryoloop and soaked with cryoprotectant solution (wild-type crystal, 30% PEG6000, 0.15 M MES (pH 6.0); S131A and DM crystal, 1.5 M ammonium sulfate; S131A and DM crystal, 0.3 M NaCl, 0.15 M Bis-Tris (pH 6.5), 20% glycerol) prior to data collection at 100 K. The diffraction images were processed using HKL2000[16]. The crystal structure of wild type PETase was solved by molecular replacement (MR) method with Phaser[17] using the structure of *Saccharomonospora viridis* cutinase-like enzyme (PDB ID code 4WFI[18], 47% sequence identity with PETase) as a search model. Further refinement was carried out by using the programs Refmac5[19] and Coot[20]. Five percent randomly selected reflections were set aside for calculating $R_{\text{free}}$ as a monitor. The variant and complex structures were determined by MR method with Phaser using the refined PETase structure as a search model. The $2F_o - F_c$ difference Fourier maps showed clear electron densities for most amino-acid residues. Subsequent refinements by incorporating ligands and water molecules were according to $1.0\sigma$ map level. The data collection and refinement statistics are summarized in Supplementary Table 2. All figures of the protein and ligand structures were prepared by using the PyMOL program (http://pymol.sourceforge. net/).

**PET modeling.** A two-unit oligomer of PET was superimposed onto the bound HEMT in the complex crystal structure based on the TPA moiety, whereas the ethylene group connecting to the other TPA moiety in the PET took the place of the hydroxyethyl group of HEMT. The position of the second TPA moiety was adjusted manually by torsional rotations of the connecting single bonds while the first TPA remained largely fixed. After eliminating most major clashes between the non-hydrogen atoms of PET and PETase, the enzyme–substrate complex model was subjected to several cycles of dynamic energy minimization by using PHENIX[21].

**Enzyme activity measurement.** Purified protein was incubated with PET film (⌀6 mm) in a buffer containing 50 mM glycine-NaOH (pH 9.0) at 30 °C for 42 h. The reaction was terminated by dilution of the aqueous solution with 160 mM phosphate buffer (pH 2.5) followed by heat treatment (85 °C, 10 min). The supernatant was obtained by centrifugation (15,000 × g, 10 min). After 0.22-µm filtration, 20 µL

of assay solution was analyzed using a high-performance liquid chromatography system (HPLC, Agilent 1200) equipped with a Welch Ultimate XB-C18 column (4.6 × 250 mm, 5 μm, Welch Materials, Inc., Shanghai, China). The mobile phase was methanol with 20 mM phosphate buffer (pH 2.5) at a flow rate of 0.8 mL min$^{-1}$, and the effluent was monitored at a wavelength of 240 nm. The typical elution condition was 0–25 min with 25–85% methanol linear gradient. The hydrolytic products of MHET and TPA were identified by comparing the retention time of standard compounds. The peak areas of MHET and TPA were calculated and presented as percentage of the wild type enzyme products. All samples were analyzed in triplicate in each independent experiment, from which the data were averaged and the standard errors were calculated.

**Data availability**. The atomic coordinates and structure factors of the reported structures have been deposited in the Protein Data Bank under accession codes as follows: wild type (WT), 5XG0; S131A, 5XFY; R103G/S131A double mutant (DM), 5XFZ; DM-HEMT, 5XH3; DM-pNP, 5XH2. All other relevant data are available from the corresponding author upon request.

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

## Acknowledgements

This work was supported by the National Natural Science Foundation of China (grants 31400678, 31470240, 31500642, 31570130, 31600058, and 31600638), CAS Interdisciplinary Innovation Team, Youth Innovation Promotion Association CAS, and by the Taiwan Protein Project (grant no. MOST105–0210–01–12–01 and MOST106-0210-01-15-04). We thank the National Synchrotron Radiation Research Center of Taiwan for beam-time allocation and data-collection assistance.

## Author contributions

X.H. and J.M. carried out cloning, protein purification, crystal growing, and enzyme activity determination. W.L., J.-W.H., and Y.-S.C. collected crystallographic data and solved all structures. Y.Z. and L.X. help purification and crystal growing of variants. T.-P.K. and C.-C.C. analyzed structural results and designed mutants. R.-T.G. designed and supervised research and prepared the manuscript with contributions from all authors.

## Additional information

**Competing interests:** The authors declare no competing financial interests.

