## [Peer Review File · Nature Communications]

Reviewers' comments:

Reviewer #1 (Remarks to the Author):

The paper describes the crystal structure and mechanism of action of a recently identified novel PET hydrolase from *Ideonella sakaiensis* (PETase). Various hydrolytic enzymes have been shown to cleave the ester bonds in the PET polymer, but their activity is rather low as PET is not a natural substrate of these enzymes. The PET hydrolase from *I. sakaiensis* is unique, as it has a natural role in PET degradation allowing the bacterial strain to use PET as a carbon source (Yoshida et al., 2016, *Science* 351, 1196). Its PET degrading activity is significantly higher compared to the other enzymes, which makes PETase highly attractive for biotechnological applications towards PET waste reduction and/or recycling. With the availability of its crystal structure, and the insights into the mechanism of substrate binding and cleavage, protein engineering efforts can now be focussed on further improving its enzymatic properties. In addition, evolutionary questions can be addressed how enzymes evolve to acquire new or improved activities.

The research described in the paper is overall sound and relatively straightforward. Crystal structures were determined of the wild-type enzyme without substrate, of two inactive mutants and of two complexes with bound ligands (substrate and product mimicks). The quality of the structures is excellent as judged from the crystallographic statistics. Mutagenesis experiments were used to analyse the importance of various residues for the PETase activity of the enzyme. The proposed overall catalytic pathway seems correct, but this is somewhat trivial as the PETase uses a catalytic triad mechanism common to many esterases and lipases. My main criticism is that the paper lacks a bit in novelty, apart from delivering structural data that is crucial for future protein engineering and/or enzyme evolutionary research. Also, I find the conclusions concerning the structural determinants of the enhanced PETase activity not very convincing. In particular I am concerned that the authors exaggerate the importance of the flexible tryptophan residue at the active site (W156) for the enhanced PETase activity of this enzyme (as compared to the other PETase-like enzymes). In my opinion there is insufficient data to make such a claim. In fact, one may just as well argue that the flexibility has a negative effect on substrate binding affinity, due to the entropic penalty effect resulting from fixing the side chain upon substrate binding. The suggestion put forward by the authors that the lower activity of the S185H mutant is due to the histidine side chain preventing flexibility of W156 could be easily tested by a reverse mutation in one of the PETase homologs, which should then show increased PET degrading activity. A more thorough structural comparison of the PETase with its structural

homologs, combined with multiple sequence analysis, should provide more information about the common and unique features of this enzyme. What are the adaptations that have increased its PETase activity? To address this question also requires a more thorough analysis of the putative binding site for a second TPA unit of the PET polymer, revealed by the docking experiments. One may expect more divergence at this region as it is further away from the active site.

In conclusion: the paper presents important data for future research, in particular towards developing bio-based technologies for PET degradation and/or recycling. However, it does not deliver new and convincing insights that explain how this enzyme has evolved towards a PET-specific hydrolase.

Reviewer #2 (Remarks to the Author):

This manuscript by Han et al. presents the crystal structure of poly(ethylene terephthalate) esterase, an enzyme that hydrolyses this plastic polymer. As such, this represents an important enzyme for the management of the ever-increasing issue of accumulating plastic wastes. An understanding of this enzyme may help with the development of an efficient biodegradation process, and thereby greatly alleviate the problem of waste management we currently face worldwide. Hence, this study has the potential of great significance. The authors solved the structure of the enzyme with relatively high resolution in complex with ligands which permitted the identification of binding and catalytic residues. The importance of some of these residues was further investigated using site-directed mutagenesis for their replacement. The manuscript has been prepared with care and is easy to read. Unfortunately however, in an attempt to make it short and concise, important information is missing which needs to be addressed. This and other issues are elaborated on below.

1. The manuscript has not been prepared as an Article following the Journal requirements for Introduction, Results and Discussion sections. Instead, it is prepared more like a letter, with text flowing through. To compound this, the results and discussion are too brief and both data and appropriate explanations are missing. As an example, the first line of the presumed Results section begins with a description of the three dimensional fold of the enzyme and that a strictly conserved catalytic triad of Ser-His-Asp is found on its surface, all in the first sentence. No further information about the structure, including crystal forms obtained, is presented which, as it turns out from later text (line 74), there are three - only two of these are referred to on line 69 of the text! This is important because the authors use this (missing) information to account for “wobbling” of a Trp residue that distinguishes it from “other” unidentified esterases (line 70). In this regard, the text on line 82 refers to only two of these crystal forms - what about the third?

Other examples of missing information in this first "results" section include how the catalytic triad was identified. Moreover, no further experimentation was conducted to confirm the identification of the catalytic triad, such as site-specific replacements and/or trapping a catalytic intermediate (eg., acetylated enzyme). The structure of the enzyme with respect to disulfide bonds was then compared to "other enzymes" (lines 59, and 68) - which other enzymes? - and that the disulfide connects the C-terminal helix to a loop - which loop?

2. The authors observed that the packing of the crystal forms of the wild-type enzyme precluded the opportunity to bind ligand; the side-chain of Arg103 sticks into the binding cleft. To circumvent this, the authors replaced the Arg with Ala, but they did this in the background of a catalytic Ser131 variant of the enzyme. Why the double mutant? Was an attempt made to prepare co-crystals of the single, Arg variant PETase without success? (From Supplementary Table 2, it appears that this enzyme variant was crystallized. This is important because the authors are then discuss details of the catalytic mechanism without being able to observe the catalytic Ser in the ligand-bound crystal structure. Also, no comparison of the two solved structures is presented - what is the extent of the superpositioning? While unlikely, the authors need to note any differences between the two structures and not simply expect none.

3. There is no mention within the text nor Methods of experiment repetition nor experimental error except in the legend to Figure 2; the latter states that measurements were made in triplicate and SD was calculated but error bars are absent. Also, the authors need to pay attention to significant figures. For example, the data presented within lines 129-141 varies between 2 and 4 significant figures.

4. Lines 143-151: The authors present a proposed mechanism of action of the esterase without any mention of water involvement, and thus ignore the second half of the reaction. The mechanism needs to be expanded.

Minor points:

1. Line 82 (and elsewhere): "mutant" should be replaced with "variant"

2. Lines 95 and 106: Do the authors mean "Pi-stacked" or T-shaped structures involving the aromatic rings?

3. Line 98: Expansion of the text justifying the identification of the putative oxyanion hole residues is required; currently the text simply refers to the main-chain amides of these two residues as The oxyanion hole.

4. Line 132: Expansion of the text concerning the participation of S185, I179, W130, and M132

is required to justify the statement here that they “might be one of the major determinants” for high activity.

5. In the description of the assay for PETase, how was microbial growth controlled/prevented during the 42 hr incubation? What was the negative control? How was quantification of the two reaction products determined? (by absorbance using a coefficient?)

6. Scheme 1 - Define EG as ethylene glycol. Also, it is not apparent to this reviewer where the carbon dioxide comes from, and what a co-product would be if indeed generated?

Reviewers' comments:

Reviewer #1 (Remarks to the Author):

The paper describes the crystal structure and mechanism of action of a recently identified novel PET hydrolase from *Ideonella sakaiensis* (PETase). Various hydrolytic enzymes have been shown to cleave the ester bonds in the PET polymer, but their activity is rather low as PET is not a natural substrate of these enzymes. The PET hydrolase from *I. sakaiensis* is unique, as it has a natural role in PET degradation allowing the bacterial strain to use PET as a carbon source (Yoshida et al., 2016, Science 351, 1196). Its PET degrading activity is significantly higher compared to the other enzymes, which makes PETase highly attractive for biotechnological applications towards PET waste reduction and/or recycling. With the availability of its crystal structure, and the insights into the mechanism of substrate binding and cleavage, protein engineering efforts can now be focused on further improving its enzymatic properties. In addition, evolutionary questions can be addressed how enzymes evolve to acquire new or improved activities.

Ans: Thanks.

The research described in the paper is overall sound and relatively straightforward. Crystal structures were determined of the wild-type enzyme without substrate, of two inactive mutants and of two complexes with bound ligands (substrate and product mimicks). The quality of the structures is excellent as judged from the crystallographic statistics. Mutagenesis experiments were used to analyse the importance of various residues for the PETase activity of the enzyme.

Ans: Thanks.

The proposed overall catalytic pathway seems correct, but this is somewhat trivial as the PETase uses a catalytic triad mechanism common to many esterases and lipases. My main criticism is that the paper lacks a bit in novelty, apart from delivering structural data that is crucial for future protein engineering and/or enzyme evolutionary research.

Ans: Although PETase, as well as other homologous PET-degrading enzymes, adopts the common catalytic triad mechanism of hydrolysis, how the enzyme binds to the substrate remains unexplored. Due to lack of substrate-bound protein complex structure, which is the most desired information, enzyme

engineering studies can only be performed by using direct evolution. As a result, no underlying mechanism for substrate binding can be provided even if any variant shows improved enzymatic activity. This greatly hampers further researches and applications. Therefore, our crystal structures which provide the first concrete evidence of substrate-binding mode of PETase shall offer valuable information for basic research and application.

Also, I find the conclusions concerning the structural determinants of the enhanced PETase activity not very convincing. In particular I am concerned that the authors exaggerate the importance of the flexible tryptophan residue at the active site (W156) for the enhanced PETase activity of this enzyme (as compared to the other PETase-like enzymes). In my opinion there is insufficient data to make such a claim. In fact, one may just as well argue that the flexibility has a negative effect on substrate binding affinity, due to the entropic penalty effect resulting from fixing the side chain upon substrate binding. The suggestion put forward by the authors that the lower activity of the S185H mutant is due to the histidine side chain preventing flexibility of W156 could be easily tested by a reverse mutation in one of the PETase homologs, which should then show increased PET degrading activity.

Ans: We thank the Reviewer for these comments. The W156 wobbling as observed in the PETase structures is unambiguous and it has not been found in other homologous structures. Structure superimposition demonstrates that the W156 in “C” conformer, which is adopted by other PET hydrolyzing enzymes, is not favorable for substrate binding (please see supplementary figure 8 in the revised manuscript). In this case, the nearby S185 that allows W156 conformational change is apparently more beneficial than a His which causes a spatial hindrance and fixes W156 in the “C” conformation in the other homologs. Our mutagenesis experiments have shown convincing evidence to support the importance of W156-S185 coupling in PETase. Nevertheless, other factors might involve in the various substrate preference that differs between PETase and the homologous enzymes. As this study mainly focuses on the analysis of PETase, which provides the first structural evidence of substrate analogs binding mode in a PET degrading enzyme, our main goal is to analyze the PETase *per se*. However, we agree with the Reviewer that these results should be presented more clearly and have rewritten the text accordingly. Furthermore, we appreciate the Reviewer’s opinions that these findings can be further strengthened by more experiments. Therefore, we will construct the suggested variants of other homologous enzymes to see if their performance

can be boosted by replacing the S185-corresponding His to Ser. In addition, we are also trying to investigate the W156 conformation of S185H variant via solving the crystal structures. The further results will go to another paper.

A more thorough structural comparison of the PETase with its structural homologs, combined with multiple sequence analysis, should provide more information about the common and unique features of this enzyme. What are the adaptations that have increased its PETase activity? To address this question also requires a more thorough analysis of the putative binding site for a second TPA unit of the PET polymer, revealed by the docking experiments. One may expect more divergence at this region as it is further away from the active site.

Ans: We are grateful for the Reviewer's suggestion and have included more active site analyses in the revised manuscript. In supplementary figure 7, the PET-interacting residues in all enzymes mentioned in the manuscript are shown. These residues are also indicated in the sequence alignment (supplementary figure 1). All of the substrate-interacting residues are strictly or semi-conserved. The second TPA unit binding site involves fewer interactions which are also consistently found in all enzymes.

In conclusion: the paper presents important data for future research, in particular towards developing bio-based technologies for PET degradation and/or recycling. However, it does not deliver new and convincing insights that explain how this enzyme has evolved towards a PET-specific hydrolase.

Ans: It is hard to explain how PETase was evolved from, if there is any, parental enzymes, as the enzyme was isolated from a novel PET-assimilating bacterium. Nevertheless, our results have disclosed the PET-binding mode of a biocatalyst for the first time, which is of pivotal importance in understanding the hydrolase at a molecular level and provides a fundamental basis for further enzyme engineering and applications. In conclusion, it is clear that our results are of great novelty and shall bring a great impact in the related research field.

Reviewer #2 (Remarks to the Author):

This manuscript by Han et al. presents the crystal structure of poly(ethylene terephthalate) esterase, and enzyme that hydrolyses this plastic polymer. As such, this represents an important enzyme for the management of the

ever-increasing issue of accumulating plastic wastes. An understanding of this enzyme may help with the development of an efficient biodegradation process, and thereby greatly alleviate the problem of waste management we currently face worldwide. Hence, this study has the potential of great significance. The authors solved the structure of the enzyme with relatively high resolution in complex with ligands which permitted the identification of binding and catalytic residues. The importance of some of these residues was further investigated using site-directed mutagenesis for their replacement. The manuscript has been prepared with care and is easy to read. Unfortunately however, in an attempt to make it short and concise, important information is missing which needs to be addressed. This and other issues are elaborated on below.

Ans: We thank the reviewer's comment and have modified the manuscript into an *Article* format which now includes more discussions and analyses.

The manuscript has not been prepared as an Article following the Journal requirements for Introduction, Results and Discussion sections. Instead, it is prepared more like a letter, with text flowing through. To compound this, the results and discussion are too brief and both data and appropriate explanations are missing. As an example, the first line of the presumed Results section begins with a description of the three dimensional fold of the enzyme and that a strictly conserved catalytic triad of Ser-His-Asp is found on its surface, all in the first sentence. No further information about the structure, including crystal forms obtained, is presented which, as it turns out from later text (line 74), there are three - only two of these are referred to on line 69 of the text! This is important because the authors use this (missing) information to account for "wobbling" of a Trp residue that distinguishes it from "other" unidentified esterases (line 70). In this regard, the text on line 82 refers to only two of these crystal forms - what about the third?

Ans: We have modified the description of crystal structure and added more information to present our results more clearly. In line 74, a sentence has been added to describe the W156 conformer in C chain: "while a Ser residue can yield a space to permit the "C" conformer of W156". In line 82, the two crystal forms mean the wild-type PETase and S131A crystals, not the "three polypeptide chains" in an asymmetric unit of wild-type PETase.

Other examples of missing information in this first "results" section include how the catalytic triad was identified. Moreover, no further experimentation

was conducted to confirm the identification of the catalytic triad, such as site-specific replacements and/or trapping a catalytic intermediate (eg., acetylated enzyme).

Ans: Identification of PETase catalytic triad is mainly based on sequence alignment with other homologous cutinases. The presence of the conserved catalytic triad and oxyanion hole is a criterion of this family of enzyme. From the crystal structure, the nucleophile S131 is within H-bond distance to the base H208, which is stabilized by the acid D177. This charge-relay network guarantees the nucleophile to be polarized and activated, and is the structural evidence of a typical catalytic triad. We also experimentally validated that S131 is essential for the enzyme activity (Fig. 2). These descriptions have now been added in the first and second paragraph of the revised manuscript for better understanding of the above-referred points.

The structure of the enzyme with respect to disulfide bonds was then compared to “other enzymes” (lines 59, and 68) - which other enzymes? - and that the disulfide connects the C-terminal helix to a loop - which loop?

Ans: The referred “the other enzymes” means the other homologous enzymes that are listed in the Supplementary figure 1, and have been modified as “homologous enzymes”. The DS2-linked loop is the “last” loop of the structures, also corrected.

2. The authors observed that the packing of the crystal forms of the wild-type enzyme precluded the opportunity to bind ligand; the side-chain of Arg103 sticks into the binding cleft. To circumvent this, the authors replaced the Arg with Ala, but they did this in the background of a catalytic Ser131 variant of the enzyme. Why the double mutant? Was an attempt made to prepare co-crystals of the single, Arg variant PETase without success? (From Supplementary Table 2, it appears that this enzyme variant was crystallized) This is important because the authors are then discuss details of the catalytic mechanism without being able to observe the catalytic Ser in the ligand-bound crystal structure. Also, no comparison of the two solved structures is presented - what is the extent of the superpositioning? While unlikely, the authors need to note any differences between the two structures and not simply expect none.

Ans: We prepared R103G/S131A double mutant crystals in an attempt to obtain ligand-bound complex structures. We also utilized R103G crystals but did not obtain any substantial electron density map of ligands. One possibility

is that the ligands were hydrolyzed by the enzyme, same as using wild type crystals. In most cases, ligands, including substrate and analogues, contain moieties that are readily attacked by catalytic residues. Thus, inactive mutant protein crystals are usually utilized to obtain intact substrate/analogue. We agree with the reviewer that inspecting the effect of mutations on structure is necessary. Superimposition of the wild type, S131A, and R103G/S131A structures reveals no significant overall structural alteration (RMSD 0.144-0.2Å), suggesting that the mutations did not affect the protein folding. This information has been added in the second paragraph of the revised manuscript.

3. There is no mention within the text nor Methods of experiment repetition nor experimental error except in the legend to Figure 2; the latter states that measurements were made in triplicate and SD was calculated but error bars are absent. Also, the authors need to pay attention to significant figures. For example, the data presented within lines 129-141 varies between 2 and 4 significant figures.

Ans: The error bars were shown in the Figure 2c but the values are too small to be clearly seen. That is why we put some of the error values in the text for better demonstration. In the revised manuscript, the error bars are presented in black color. But overall it indicates that the activity measurements were conducted with good reproducibility. Also, we have added triplicate measurement and error calculation in the Method section in the revised manuscript for clarity. The significant figures in the results section have been unified.

4. Lines 143-151: The authors present a proposed mechanism of action of the esterase without any mention of water involvement, and thus ignore the second half of the reaction. The mechanism needs to be expanded.

Ans: The reaction mechanism of cutinases (EC 3.1.1.74), a member of serine hydrolase superfamily, is well known. PETase that is equipped with a classical Ser-His-Asp catalytic triad is expected to adopt the typical mechanism of cutinases which comprises two-stage nucleophilic attack. The first one is carried out by the nucleophile Ser and the second one is by a water molecule. What remain to be explored for PET-hydrolytic enzymes is how the substrate-enzyme interaction takes place and which residues are involved. The mechanism figure depicts this hitherto unknown part, as revealed by the new insights provided in the present study, but omits the remaining well-known

process. We thank the reviewer for the suggestion and have instead added previous review articles of cutinase/esterase mechanism of action to provide complete information of PETase catalysis.

Minor points:

1. Line 82 (and elsewhere): “mutant” should be replaced with “variant”

Ans: Corrected.

2. Lines 95 and 106: Do the authors mean “Pi-stacked” or T-shaaped structures involving the aromatic rings?

Ans: T-stacking force means the T-shaped conformation.

3. Line 98: Expansion of the text justifying the identification of the putative oxyanion hole residues is required; currently the text simply refers to the main-chain amides of these two residues as The oxyanion hole.

Ans: Oxyanion hole is a common feature of cutinases and other members in the hydrolase superfamily. To further clarify this point, additional descriptions of the oxyanion hole along with a cutinase review have been included in the referred text.

4. Line 132: Expansion of the text concerning the participation of S185, I179, W130, and M132 is required to justify the statement here that they “might be one of the major determinants” for high activity.

Ans: We appreciate the reviewer’s suggestion and more descriptions have been added to the referred section.

5. In the description of the assay for PETase, how was microbial growth controlled/prevented during the 42 hr incubation? What was the negative control? How was quantification of the two reaction products determined? (by absorbance using a coefficient?)

Ans: As the Reviewer mentioned, because the enzyme reaction of PETase was incubated at 30 °C for as long as 42 hours, we had to carefully control microbial contamination. All reagents including the protein solution were passed through 0.22 µm filter and all vessels and tips were sterilized. The entire operation procedure was conducted in a lamina flow cabinet.

The negative control of reaction is set up by using the same reaction system

without adding the enzyme. The retention time for TPA and MHET were 13.7 min and 14.6 min respectively, which were determined by comparing to the standard samples. The peak areas of two reaction products were measured and presented by percentage of the wild type enzyme products. All samples were analyzed in triplicate in each independent experiment, and standard errors were calculated. These descriptions were now added in the revised manuscript.

6. Scheme 1 - Define EG as ethylene glycol. Also, it is not apparent to this reviewer where the carbon dioxide comes from, and what a co-product would be if indeed generated?

Ans: We thank the Reviewer for this comment and have modified scheme 1 as requested. Carbon dioxide is not present in the reaction and it has been removed from the figure.

** See Nature Research's author and referees' website at www.nature.com/authors for information about policies, services and author benefits

This email has been sent through the Springer Nature Tracking System NY-610A-NPG&MTS

Confidentiality Statement:

This e-mail is confidential and subject to copyright. Any unauthorised use or disclosure of its contents is prohibited. If you have received this email in error please notify our Manuscript Tracking System Helpdesk team at <http://platformsupport.nature.com>.

Details of the confidentiality and pre-publicity policy may be found here <http://www.nature.com/authors/policies/confidentiality.html>

Privacy Policy | Update Profile

Reviewers' Comments:

Reviewer #1 (Remarks to the Author):

With the revised version of their manuscript, the authors adequately addressed most of the points raised by me and the other reviewer. The structural basis for the enhanced PET hydrolase activity exhibited by the *I. sakaiensis* enzyme remains somewhat unclear, though, including the role of the “wobbling” tryptophan, but future research may shed more light on this. Considering the high interest of this enzyme for its catalytic, evolutionary and biotechnological properties, and the potential importance of the results for future plastic waste remediation, I recommend publication of this paper.

A few extra minor revisions to the main text are necessary to further improve the paper:

1) Line 24: “researches” —> research

2) Line 65: “solved at 1.58 Å” —> solved at 1.58 Å resolution

3) Line 68: “... homologous enzymes (Supplementary Fig. 1)”: the authors should specify that the set of homologous enzymes (according to suppl. fig. 1) is restricted to the ones exhibiting PET-hydrolytic activity, or provide another reference (if the actually chosen set of homologous enzymes is larger)

4) Line 72: “... while most other homologous enzymes have only one (Supplementary Figs. 1 and 2).”: see previous remark. Actually, in the chosen set of homologous enzymes (with PET-hydrolytic activity) all other enzymes have only one disulphide bond.

5) Line 102: “ ... the R103 side chain of one protein ...” —> the R103 side chain of one protein molecule

6) Line 103: “ ... of another protein ... “ —> of another protein molecule

7) Line 109: “RMSD”: write out the full name. Also, specify which parts of the structures were used for the overlay (I assume C α -backbones?)

8) Line 131-132: “Therefore, PETase might exploit similar substrate-interaction network as seen in the other enzymes”: this is a somewhat confusing conclusion. Has the “substrate-interaction” network actually been observed in the other enzymes or is it inferred from the conservation of substrate-interacting residues in PETase? If the first statement is true, then the authors should

provide references to the published work. If the latter is true, than the authors should rephrase this sentence. My suggestion (assuming that the latter statement is true): “Thus it appears that PETase and the other enzymes with PET-hydrolyzing activity use a similar substrate-interaction network”

9) Line 142: “The ligand of DM-pNP is located to the same side as HEMT”: the authors should rephrase this sentence. What is meant with “to the same side”?

10) Line 166-167: “Notably, replacing W130 with a His, which is consistently observed in other homologous enzymes, significantly reduced the PETase activity”. Can the authors provide an explanation for this significant reduction in activity, which is much more dramatic than the effect of the S185H mutation?

11) Lines 116, 150, 183, 184, Figure 3: The authors use the terms T-stacked and π -stacked (or π -stacking interaction) to distinguish between two different geometries of aromatic interactions. The term “ π -stacked” is misleading though, as T-stacked is also a form of π -stacking. A better term for the interaction observed between the phenyl ring of pNP and W156 would be “face-to-face stacked” or “parallel displaced π -stacking”

Reviewer #2 (Remarks to the Author):

The authors have adequately addressed each of the concerns raised in my original review. To be more clear, the issues regarding details on the mechanism of action were raised recognising the broad readership of Journal where many readers may not have the appropriate background in enzyme structure and function.

REVIEWERS' COMMENTS:

Reviewer #1 (Remarks to the Author):

With the revised version of their manuscript, the authors adequately addressed most of the points raised by me and the other reviewer. The structural basis for the enhanced PET hydrolase activity exhibited by the *I. sakaiensis* enzyme remains somewhat unclear, though, including the role of the “wobbling” tryptophan, but future research may shed more light on this. Considering the high interest of this enzyme for its catalytic, evolutionary and biotechnological properties, and the potential importance of the results for future plastic waste remediation, I recommend publication of this paper.

Response: thank you.

A few extra minor revisions to the main text are necessary to further improve the paper:

1) Line 24: “researches” → research

Response: deleted.

2) Line 65: “solved at 1.58 Å” → solved at 1.58 Å resolution

Response: corrected.

3) Line 68: “... homologous enzymes (Supplementary Fig. 1)”: the authors should specify that the set of homologous enzymes (according to suppl. fig. 1) is restricted to the ones exhibiting PET-hydrolytic activity, or provide another reference (if the actually chosen set of homologous enzymes is larger)

Response: corrected.

4) Line 72: “... while most other homologous enzymes have only one (Supplementary Figs. 1 and 2).”: see previous remark. Actually, in the chosen set of homologous enzymes (with PET-hydrolytic activity) all other enzymes have only one disulphide bond.

Response: the reviewer is correct and the “most” has been deleted.

5) Line 102: “... the R103 side chain of one protein ...” → the R103 side chain of one protein molecule

Response: corrected.

6) Line 103: “... of another protein ... “ → of another protein molecule

Response: corrected.

7) Line 109: "RMSD": write out the full name. Also, specify which parts of the structures were used for the overlay (I assume C α -backbones?)

Response: the reviewer is correct and the referred part has been modified to C α root mean square deviation (RMSD).

8) Line 131-132: "Therefore, PETase might exploit similar substrate-interaction network as seen in the other enzymes": this is a somewhat confusing conclusion. Has the "substrate-interaction" network actually been observed in the other enzymes or is it inferred from the conservation of substrate-interacting residues in PETase? If the first statement is true, then the authors should provide references to the published work. If the latter is true, than the authors should rephrase this sentence. My suggestion (assuming that the latter statement is true): "Thus it appears that PETase and the other enzymes with PET-hydrolyzing activity use a similar substrate-interaction network"

Response: the reviewer is correct (the "latter statement") and the sentence has been modified as the reviewer suggested.

9) Line 142: "The ligand of DM-pNP is located to the same side as HEMT": the authors should rephrase this sentence. What is meant with "to the same side"?

Response: the sentence has been modified as "The ligand of DM-pNP is located to the 1st benzene ring-binding site, the same side as HEMT."

10) Line 166-167: "Notably, replacing W130 with a His, which is consistently observed in other homologous enzymes, significantly reduced the PETase activity". Can the authors provide an explanation for this significant reduction in activity, which is much more dramatic than the effect of the S185H mutation?

Response: we thank the reviewer's comment which is interesting and may not be properly answered if no structural information is available. We shall carry out further investigation to get insight into this issue.

11) Lines 116, 150, 183, 184, Figure 3: The authors use the terms T-stacked and π -stacked (or π -stacking interaction) to distinguish between two different geometries of aromatic interactions. The term " π -stacked" is misleading though, as T-stacked is also a form of π -stacking. A better term for the interaction observed between the phenyl ring of pNP and W156 would be "face-to-face stacked" or "parallel displaced π -stacking"

Response: we thank the reviewer's comment and have modified π -stacked as "face-to-face stacked"

Reviewer #2 (Remarks to the Author):

The authors have adequately addressed each of the concerns raised in my original review. To be more clear, the issues regarding details on the mechanism of action were raised recognising the broad readership of Journal where many readers may not have the appropriate background in enzyme structure and function.

Response: thanks.